# An Effective Learning Method for Automatic Speech Recognition in Korean CI Patients' Speech

**Jiho Jeong [1], S. I. M. M. Raton Mondol [1] , Yeon Wook Kim [2] and Sangmin Lee [1,2,*]**

[1] Department of Electrical and Computer Engineering, Inha University, Incheon 22212, Korea;
fd873630@naver.com (J.J.); simmraton@gmail.com (S.I.M.M.R.M.)
[2] Department of Smart Engineering Program in Biomedical Science & Engineering, Inha University,
Incheon 22212, Korea; kimywih1@naver.com
* Correspondence: sanglee@inha.ac.kr

**Abstract:** The automatic speech recognition (ASR) model usually requires a large amount of training data to provide better results compared with the ASR models trained with a small amount of training data. It is difficult to apply the ASR model to non-standard speech such as that of cochlear implant (CI) patients, owing to privacy concerns or difficulty of access. In this paper, an effective finetuning and augmentation ASR model is proposed. Experiments compare the character error rate (CER) after training the ASR model with the basic and the proposed method. The proposed method achieved a CER of 36.03% on the CI patient's speech test dataset using only 2 h and 30 min of training data, which is a 62% improvement over the basic method.

**Keywords:** speech recognition; finetuning; CI patients; augmentation

## 1. Introduction

Various automatic speech recognition (ASR) models have been proposed in recent years, including the recurrent neural network transducer (RNN-T) [1], Listen, Attend and Spell (LAS) [2], and Deep Speech 2 [3]. ASR models trained on a large training dataset have achieved excellent results and are used in personal phones, IoT (Internet of Things) devices, and cloud services, examples of which include Alexa, Siri, and Bixby. However, non-standard speech, such as amyotrophic lateral sclerosis (ALS) speech, Parkinson's speech, and cochlear implant (CI) patients' speech, has a low recognition rate, whereas ASR models are trained using standard speech data sets [4,5]. Therefore, people with non-standard speech cannot use ASR models trained with a standard speech dataset.

People with severe hearing loss have low speech understanding even if they use hearing aids. Therefore, speech understanding can be restored through cochlear implants in people with severe hearing loss, especially sensorineural hearing loss. However, CI involves some trade-offs because the patient's residual acoustic hearing will no longer be available and only electrical stimulation is possible. For example, the mean spectral energy of CI patients is lower than that of normal individuals, implying that they have difficulty in pronouncing high-frequency words. Patients with CI also pause (staccato) in the middle of the speech and often confuse voiceless with voiced speech [6,7]. The characteristics of CI patients' speech differ from those of standard speech. In this study, we experimented with an effective ASR method to increase the recognition rate of the non-standard CI patients' speech.

The biggest hindrance in learning a non-standard dataset is to find sufficient data to train an ASR model [8]. Moreover, a smaller dataset often suffers from overfitting. When overfitting occurs, an algorithm shows higher accuracy with training data but does not work properly in test data. Adversarial training can be used to generate training data by transforming standard speech into nonstandard speech [9]. However, this method requires

a clinician's knowledge to verify the transformed speech samples, which is time consuming and impractical.

In this paper, we proposed a training method consisting of two steps to solve the overfitting problem under smaller dataset conditions. First, we pre-trained the ASR model with 600 h of Korean standard speech as shown in [8], with 1000 h of English standard speech. The main difference between these two approaches is using the different dataset (English and Korean). Second, we used a data augmentation technique and selected the augmentation method [10,11] used for standard speech. To the best of our knowledge, there is no previous study on whether data augment could be applied to non-standard speech dataset. Thus, we personalized the pretrain model according to the CI patient's speech.

In Section 2, we present the base ASR model, RNN-T, and explain the learning method. Section 3 verifies the experimental presented in Section 2.

## 2. Materials and Methods

Section 2.1 describes the RNN-T [12] model that enables streaming ASR, Section 2.2 describes the pre-train process, and Section 2.3 describes three finetuning methods: Basic finetuning (method 1), Fixed Decoder (method 2) and Data Augmentation (method 3) in the speech of CI patients. Figure 1 illustrates the sequence of the experiments. Figure 1a shows the steps of training procedure and Figure 1a shows the test procedure, respectively. In Figure 1a, firstly, the base model is pre-trained with the data of standard speech. Second, the pre-train model is finetuned with non-standard train speech. Three finetuned models are derived through training with non-standard speech depending on the method (method 1, 2, 3). In test procedure, we evaluate the performance of the finetuned models with non-standard test speech.

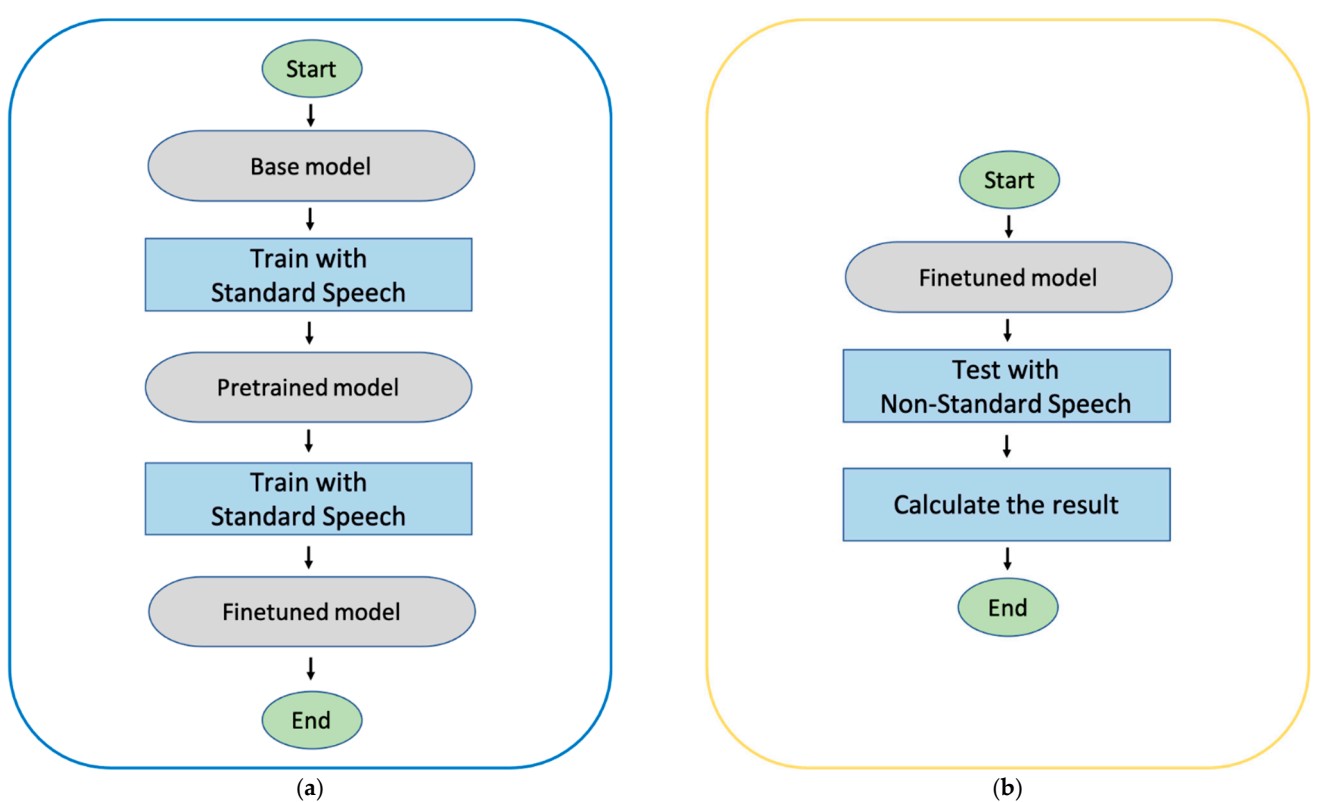

**Figure 1.** Block diagram of the experiments. (**a**) Training process. (**b**) Testing process

### 2.1. Base Model

This study uses a unidirectional RNN transducer as a base model (Figure 2). We used the version presented in [13]. The RNN-T model consists of an encoder network, a decoder network, and a joint network. Intuitively, encoder networks can be considered as acoustic models, and decoder networks are similar to language models.

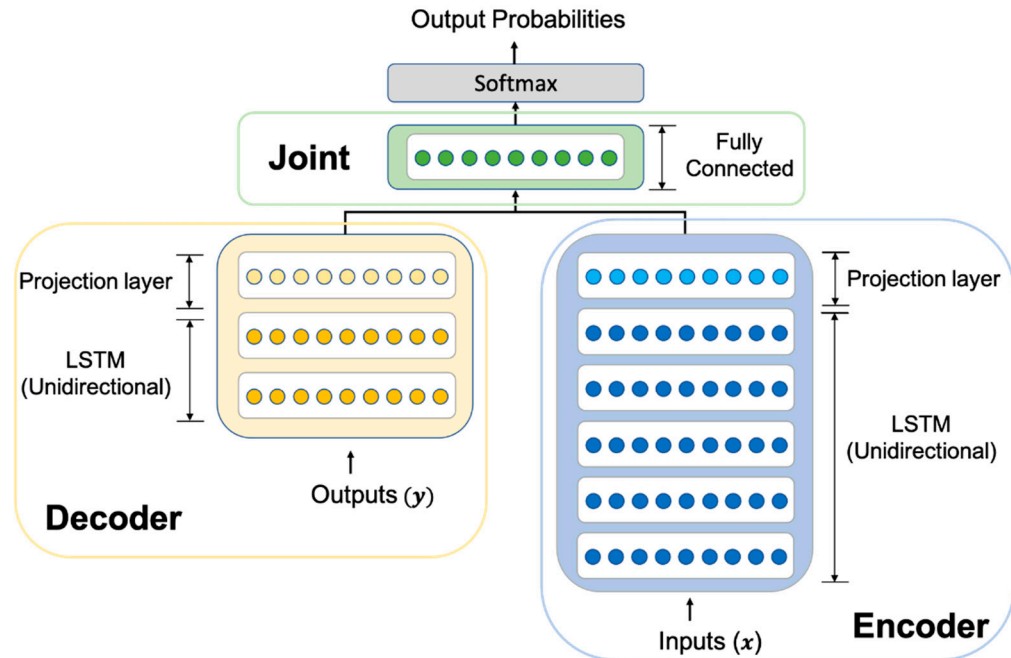

**Figure 2.** Architecture of the recurrent neural network transducer (RNN-T).

All the experiments used 80-dimensional log-Mel features, computed with a 20 ms window which was a hamming window, and shifted every 10 ms.

$x = (x_1, x_2, \ldots, x_T)$ is the input sequence of length $T$. $y = (y_1, y_2, \ldots, y_U)$ is the output sequence of length $U$. Set $Y$ consists of a total of 53 Korean onset, nucleus, and coda. For e.g., [ㄱ, ㄴ, ㄷ, ... , ㅏ, ㅑ, ㅓ, ㅕ, ... , ㅎ, ㅄ, "space"]. $y_{u-1} \in Y \cup [< sos >]$, where $< sos >$ is a special label indicating the beginning of a sentence. Both $x_t$ and $y_u$ are fixed-length real-valued vectors. In the encoder network, the input sequence is used to calculate the output vector $f = (f_1, f_2, \ldots, f_T)$ through the unidirectional RNN and projection layer. In the decoder network, the previous label prediction $(y_0, y_1, y_2, \ldots, y_U)$ calculates the output vector $g = (g_1, g_2, \ldots, g_{U+1})$ through the unidirectional RNN and projection layer, where $y_0$ is $< sos >$.

A joint network calculates the output vector $h_{t,u}$ by concatenating $h_t$ and $g_u$:

$$h_{t,u} = Joint\ Network(concat[f_t,\ g_u]) \tag{1}$$

where $h_{t,u}$ obtained through the joint network defines the probability distribution through the softmax layer.

Our model's encoder network consists of five unidirectional layers of 512 LSTM cells and 320 projection layers. The decoder network consists of two layers of 512 LSTM cells and 320 projection layers. The joint network consists of 320-feedforward layers and has 13.5 M parameters in total.

### 2.2. Pre-Train Process

The work of Shor et al. [8] was pre-trained with the Librispeech [14] dataset, an open-source dataset of 1000 h, while we pretrained with the AI hub's Korean open-source dataset, KsponSpeech [15]. KsponSpeech audio files have a format of 16 KHz/16 bits of sample/bit rate. The learning rate was set to $10^{-4}$ using the Adam optimizer [16].

### 2.3. Finetuning

This section deals with finetuning the RNN-T model to the CI patient's speech, which is pre-trained with standard speech. Sections 2.3.1 and 2.3.2 propose efficient finetuning methods that are more effective than basic finetuning.

#### 2.3.1. Fixed Decoder

Intuitively, the structure of the RNN-T can be considered an acoustic model, while the decoder network can be thought of as a language model. In this study, we used the fixed decoder method proposed in [8], which does not train decoder networks corresponding to language models within the RNN-T model structure.

#### 2.3.2. Data Augmentation

An augmentation method was used to solve the overfitting problem. Data augmentation can enhance the model performance by preventing overfitting. The augmentations used in this study were pitch perturbation, speed perturbation [10], and SpecAugment [11], all of which achieved good performance in ASR.

#### Pitch Perturbation

The spectral mean of CI patients is lower than that of normal individuals. These characteristics are used to perform pitch perturbation. Figure 3 shows examples of the individual pitch perturbation applied to a single input. The pitch factor range is between −1 and 1. The mean spectral energy of CI patient is 50 Hz higher than standard speech or 150 Hz lower than standard speech, thus determining the factor from −1 to 1. A number within the pitch factor range for each iteration was randomly selected. We modified the pitch using the pitch shift function of the librosa function.

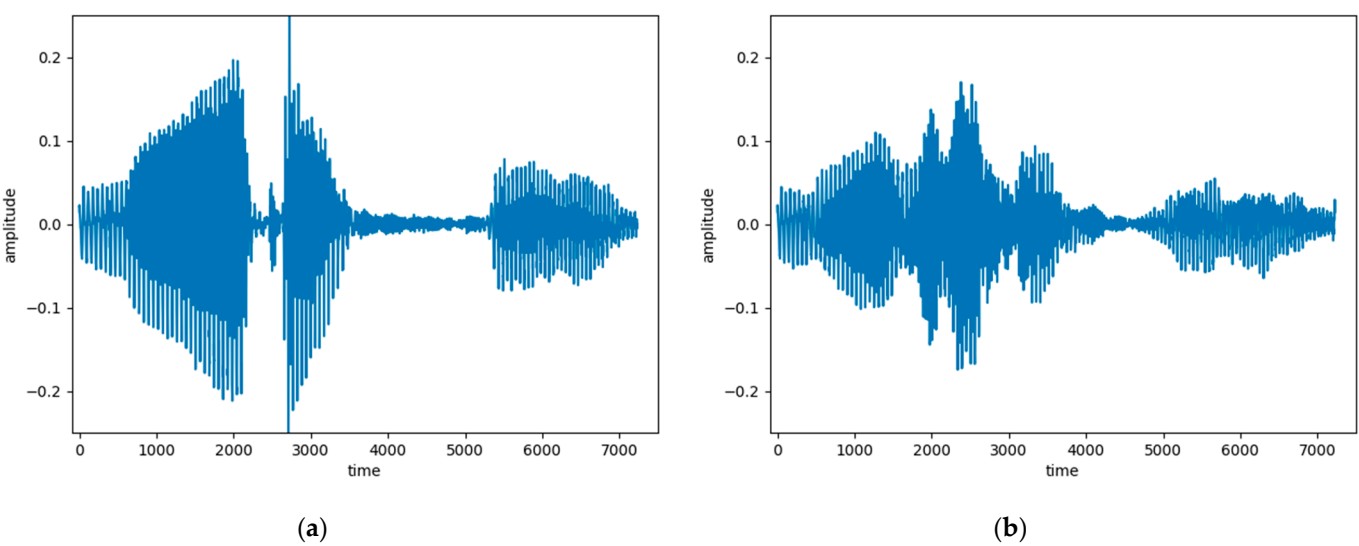

(**a**)             (**b**)

**Figure 3.** (**a**) Original input legend. (**b**) Speech signal after pitch perturbation.

#### Speed Perturbation

Refence [10] created two additional original training data by modifying the speed to 90% and 110% of the original speed. In our study, we randomly selected the speed factor as a number between 0.9 and 1.1 for each iteration based on [10]. Figure 4 shows examples of the individual speed perturbation applied to a single input. We modified the speed using the time-stretch function of the librosa function.

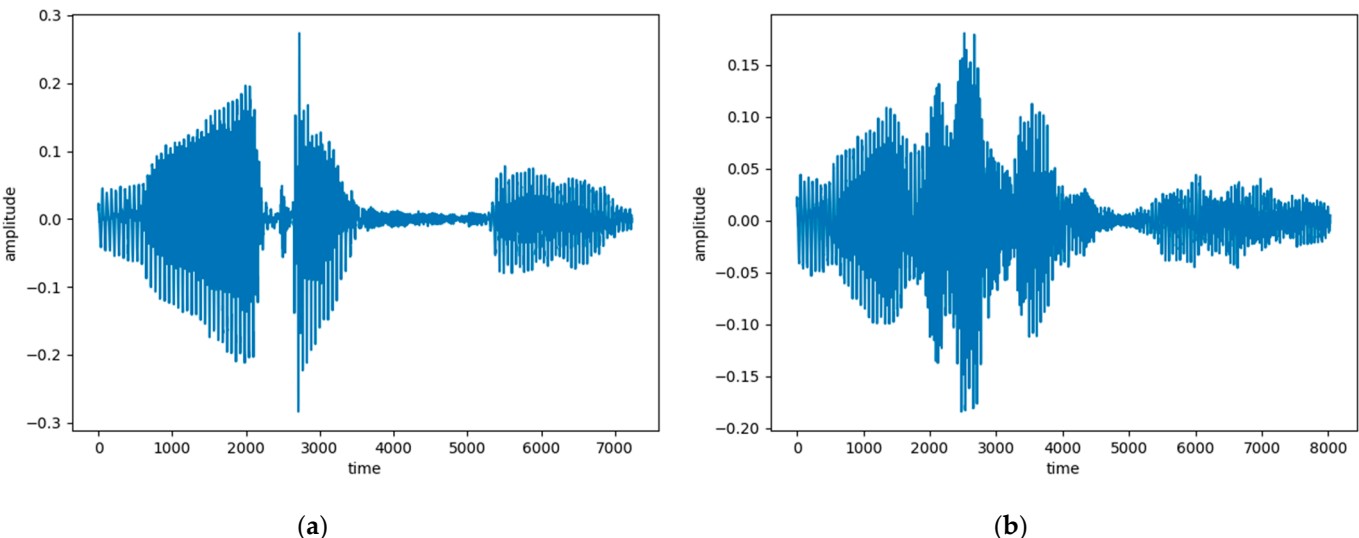

(**a**)                    (**b**)

**Figure 4.** (**a**) Original input legend. (**b**) Speech signal after speed perturbation.

Spec Augmentation

Speed perturbation and pitch perturbation were applied to the raw audio. Unlike the modification of the raw audio, SpecAugment is a method that operates on the log-Mel spectrogram of the input audio. We used time masking and frequency masking modified from "cutout" [17] proposed in the field of computer vision. The log-Mel spectrograms were normalized to have zero mean value, and thus setting the masked value to zero is equivalent to setting it to the mean value. Time warping was not performed. This method does not distort the characteristics and is simple and computationally inexpensive to apply. Figure 5 shows examples of the individual augmentations applied to a single input.

1.   Frequency masking was performed so that the frequency channels $[f_0, f_0 + F)$ were masked, where $F$ is a frequency mask parameter, and $f_0$ is selected from $[0, v - F]$, where v is the number of log-Mel channels.

2.   Time masking was performed so that the frequency channels $[m_0, m_0 + T)$ were masked, where $T$ is a time mask parameter, and $m_0$ is selected from $[0, t - T]$, where $t$ is the length of the log-Mel frequency.

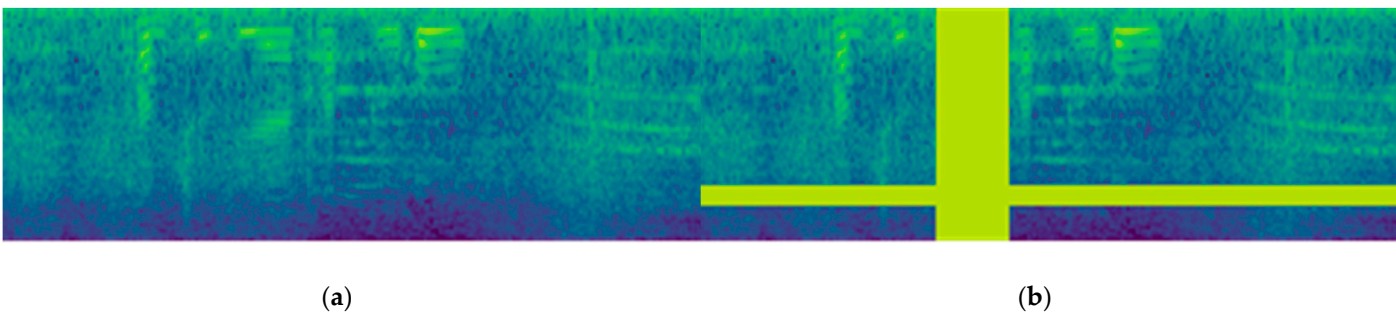

(**a**)                    (**b**)

**Figure 5.** (**a**) Spectrogram of original input. (**b**) SpecAugment applied to the input. The x-axis is the time axis, the y-axis is the frequency axis, and the color is the intensity.

### 2.4. Dataset & Metric

In 2018, the AI Hub released 1000 h of the Korean dialog dataset, KsponSpeech [14]. We pretrained our model with approximately 600 h of speech (a total of 540,000 speech utterances within a 10 s timeframe from KsponSpeech). Phonometrically, in many cases, the Korean alphabet is made up of onset, nucleus, and coda, so one phoneme is expressed differently depending on the location. Among general sentences, the frequency of consonants

and vowels in Korean speech is much different than English. In addition, while English sentences are right branching language as subject-verb-object structure, Korean sentences are left branching language as subject-object-verb structure. Therefore, Korean speech has many different characteristics than English in sentence structure and vocalization [18]. CI data was collected for about 3 h from 15 CI patients on YouTube. The speech was recorded through the subjects' cell phones and then uploaded to YouTube. The training data and evaluation data were divided into 2 h and 30 min, respectively.

We use the character error rate (CER) as a metric:

$$CER(\%) = \frac{D}{L} \times 100, \quad D = Distance_{LEV}(X, Y) \tag{2}$$

where $X$ and $Y$ are predicted, and represent the ground truth scripts. The distance $D$ is the Levenshtein distance between $X$ and $Y$, and the length $L$ is the length of the ground truth script $Y$.

### 3. Results & Discussion

Section 3.1 compares the performance of pre-train in Section 2.2, and Sections 3.2 and 3.3 compares the finetuning method of Section 2.3.

#### 3.1. Method 1 Result: The Pre-Train Process

Our first experiment was to evaluate the performance of the pre-train process. Table 1 compares a CI patient's speech model (E1) and the finetuning model (E2) after pre-training. Model E2 was pre-trained with standard speech and finetuned with a CI patient's speech. The table shows that the model pre-trained with standard speech performed better. The pre-train method stabilizes the learning of insufficient datasets.

**Table 1.** Comparison of character error rate (CER) with and without pretrain.

| Exp | RNN-T | |
|---|---|---|
| - | **Standard-Speech** [1] | **CI-Speech** [2] |
| E1 | - | 98.40 |
| E2 | 18.09 | 42.13 |

[1] Standard speech test dataset [14]. [2] CI patient's speech test dataset.

#### 3.2. Method 2 Result: Fixed Decoder without Augmentation

Section 3.1 shows that the CER improved after pre-training. However, it still demonstrated overfitting with insufficient datasets. Table 2 compares the finetuning model (E2) of all the models and that of only the encoder and the joint network (E3). It is improved by fixing the decoder network corresponding to the language model.

**Table 2.** Comparison of CER between basic finetuning and finetuning with a fixed decoder.

| Exp | RNN-T |
|---|---|
| - | **CI-Speech** |
| E2 | 42.13 |
| E3 | 37.35 |

#### 3.3. Method 3 Resul: Fixed Decoder with Augmentation

Table 3 compares the performance of the pitch perturbation (E4) and speed perturbation (E5) described in Section 2.3.2. As shown, because the CI patient's speech is already distorted, it can be confirmed that the augmentation that modifies the raw audio is not effective.

**Table 3.** Comparison of CER between speed perturbation and pitch perturbation.

| Exp | RNN-T |
|-----|-------|
| - | CI-Speech |
| E4 | 41.33 |
| E5 | 44.86 |

Table 4 shows the result of SpecAugment described in Section 2.3.2. Here, experiments (E6, E7, E8 and E9) confirmed that the SpecAugment masking the log-Mel spectrogram is more effective than the augmentation modifying the raw audio.

**Table 4.** Comparison of CER for various policies of SpecAugment. $m_F$ and $m_T$ denote the number of frequency and time masks applied, respectively.

| Exp | SpecAugment Policy | | | | RNN-T |
|-----|-------|-------|-------|-------|-------|
| - | $m_F$ | F | $m_T$ | T | CI-Speech |
| E6 | 1 | 15 | 1 | 50 | 37.23 |
| E7 | 1 | 15 | 2 | 25 | 36.07 |
| E8 | 2 | 7 | 2 | 25 | 36.03 |
| E9 | 1 | 7 | 1 | 25 | 36.41 |

*3.4. Comparison of the Results of the Three Methods*

Table 5 compares the performances of the three methods presented in this study. The finetuning method proposed in [8] to train ALS patients' speech to the ASR model has the same effect on the speech of CI patients. In addition, it was confirmed that the [11] method for augmentation of standard speech is effective for the speech of CI patients. The final result, E8, shows a 62% improvement over E1 learned with only the CI patients' speech.

**Table 5.** Comparison of CER for methods 1, 2, and 3.

| Exp | RNN-T |
|-----|-------|
| - | CI-Speech |
| E2(Method 1) | 42.13 |
| E3(Method 2) | 37.35 |
| E8(Method 3) | 36.03 |

Table 6 compares the results of previous study as in [10] with the results of our experiments. The data in the previous paper were experimented with ALS speech, which is English, but our paper was experimented with CI speech, which is Korean. Although the dataset is different, it was possible to confirm that it could be improved through the finetuning method. Furthermore, the learning data from the previous paper was 36 h, but our work was trained with limited data of 4 h. We confirm that these problems can be improved through data augmentation.

**Table 6.** Comparison of performance for method 3 and Personalizing ASR for Dysarthric and Accented Speech with Limited Data.

| Model | CER (%) | | WER (Word Error Rate) (%) | |
|-------|---------|-----------|---------|-----|
| - | Base | CI-Speech | Base | ALS |
| RNN-T (Method 3) | 98.40 | 36.03 | - | - |
| RNN-T [8] | - | - | 59.7 | 20.9 |

## 4. Conclusions

In this study, we proposed a finetuning method that can effectively train CI patients' speech. We were able to train the speech of a few of CI patients' by applying pre-train to a standard speech dataset. We did not train the decoder corresponding to the language model for the RNN-T architecture, thus achieving better performance. This claim is not precise, but the fixed decoder method can prevent overfitting for insufficient scripts. In addition, performance can be improved using SpecAugment from among the various data augmentation methods. SpecAugment does not distort raw audio, so it can improve the performance of CI patients' speech, in cases where the raw audio is already distorted. The proposed finetuning method achieves a 62% CER improvement over the other basic methods. In future, we plan to continuously improve the recognition rate for non-standard speech by studying models and methods that are robust to non-standard speech.

**Author Contributions:** Conceptualization, J.J., S.I.M.M.R.M. and S.L.; methodology, J.J., S.I.M.M.R.M. and S.L.; software, J.J.; validation, J.J. and Y.W.K.; formal analysis, J.J., Y.W.K. and S.I.M.M.R.M.; investigation, J.J., S.I.M.M.R.M. and S.L.; resources, S.L.; data curation, J.J. and Y.W.K.; writing—original draft preparation, J.J., S.I.M.M.R.M. and S.L.; writing—review and editing, J.J., S.I.M.M.R.M. and S.L.; visualization, J.J. and Y.W.K.; supervision, S.L.; project administration, S.L.; funding acquisition, S.L. All authors have read and agreed to the published version of the manuscript.

**Funding:** This research was supported by Basic Science Research Program through the National Research Foundation of Korea (NRF) funded by the Ministry of Science and ICT (NRF-2020R1A2C2004624) and by the Ministry of Education (No. 2018R1A6A1A03025523).

**Conflicts of Interest:** The authors declare no conflict of interest.

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
