# Peer review of "An Effective Learning Method for Automatic Speech Recognition in Korean CI Patients’ Speech"

_electronics, doi:10.3390/electronics10070807_

Round 1
Reviewer 1 Report
An Effective Learning Method for Automatic Speech Recognition in Korean CI Patients’ Speech
In this paper the authors present a method for improving learning in AST for CI patients.
An interesting topic in an important area. The paper is well organized and written with good english.
The introduction could provide a better description of CI speech and what are the implications in ASR. A very brief explanation of Korean ASR models would be also important (explaining symbols, phonetics and word forming. I believe that the authors can provide 2 or 3 sentences covering theses aspects). The explanation of previous work involving data augmentation could also be better explored.
The methodology is interesting and the evaluation follow a valid scientific approach.
Testing scenarios could be presented.
Comparison with other authors using similar databases or systems would improve discussion.
In conclusion, the authors could better explain what are the main contribution of the paper and what is novel.
Some additional comments/suggestions:
L1. include intro about CI speech
L27. include meaning of CI (non-mentioned in the main content)
L30. "spectral mean" -> mean spectral content or the mean spectral energy
L38. Please explain the overfitting problem
L43. "as presented in [9]". please include the name of the technique or something more specific
L44. "an ASR model was pre-trained with 1000h of standard....to learn ALS". How?
L64. 80 dimensional log-Mel ? Why 80?
L65. "harming window" ?
L72 "the input sequence ? calculates" -> "the input sequence is used to calculate"
L84. "Paer [9]" -> "In the work of Shor et al. [9]..."
L95. "which does not train decoder" -> so what does it do?
L104. between -1 and 1, please explain the use of these factors.
L108 Fig 3. Legend could better explain what is being shown. The y range should be the same. Maybe spectrograms with pitch tracking (like those generated by Praat) could help to visualize the results that are being presented. x-label doesn't shown units (is it ms?)
L110. Speed perturbation while maintaining F0 ?
Fig 4. Legend (b) Speech signal after speed perturbation
L125. F and T must be better explained.
Fig 5. What value is chosen to fill the spectrum (what is the value of the green areas in Fig 5b?)
L139. In Korean, is there a correspondence between phonemes and characters? Maybe something could be added about this. In many languages there are complex grapheme/phoneme correspondence modules.
L141. Create an introduction first where Methods 1, 2 and 3 are briefly explained
Tables 1, 2 and 3 could be combined for easier comparison
Reviewer 2 Report
Figure 1 needs to be redrawn. The same blocks are plotted twice.
Figures 2 to 5 are not mentioned.
The caption of Table 5 is comparison for three methods, but in the table other two methods are compared.
For the references, "et al" should be replaced by a full list of authors.
Round 2
Reviewer 1 Report
The authors have answered my questions in general.
Author Response
Dear Reviewer1,
Thank you for your suggestion.
We modified the reference as the guide of the journal and did spell check.
Figure 1 and description were modified.
Several typos have been fixed.
Thank you very much.
Sangmin Lee
Reviewer 2 Report
The revised Figure 1 has some problem. The two part can be combined into one, but two flows (one for training and the other for testing). They share the same fine-tuned model. For the testing flow, non-standard test speech is applied on the fine-tine model, not directly on the result.
The format of the author list in the references is inconsistent.
Author Response
Dear Reviewer2,
Thank you very much for your suggestion.
We modified our paper as you mentioned.
The detail contents are attached as a MS file.
Thank you very much.
Sangmin Lee
